# The Effectiveness of CFRP- and Auxetic Fabric-Strengthened Brick Masonry under Axial Compression: A Numerical Investigation

**DOI:** 10.3390/polym14091800

**Published:** 2022-04-28

**Authors:** Mohammad Asad, Tatheer Zahra, Julian Thamboo

**Affiliations:** 1School of Civil and Environmental Engineering, Queensland University of Technology, Brisbane 4000, Australia; m.asad@qut.edu.au; 2Department of Civil Engineering, South Eastern University of Sri Lanka, Oluvil 32360, Sri Lanka; jathamboo@seu.ac.lk

**Keywords:** auxetic fabric, bonded brickwork, CFRP, compressive strength, cyclic compression, simplified micro-modelling

## Abstract

Bonded brickwork used for loadbearing walls is widely found in heritage structures worldwide. The evaluation of bonded masonry structures and their strengthening strategies against dynamic actions require appropriate understanding under cyclic loading. Subsequently, a simplified 3D microscale numerical model is developed in this paper to analyse bonded brickwork under cyclic compression. A plasticity-based damage constitutive model to represent damage in masonry bricks under cyclic compression loading was employed, and zero-thickness interfaces were considered with non-linear damage properties to simulate the mechanical behaviour of masonry. A threshold strain level was used to enact the element deletion technique for initiating brittle crack opening in the masonry units. The developed model was validated against the experimental results published by the authors in the past. The models were able to accurately predict the experimental results with an error limit of 10% maximum. Mainly, two types of strengthening materials, possessing (1) high energy absorption characteristics (auxetic fabric) and (2) high strength properties (carbon fibre reinforced polymer composites/CFRP) were employed for damage mitigation under cyclic compression. Results show that the CFRP-strengthened masonry failure was mainly attributed to de-bonding of the CFRP and crushing under compression. However, the auxetic strengthening is shown to significantly minimise the de-bonding phenomenon. Enhanced energy dissipation characteristics with relatively higher ductility (up to ~50%) and reduced damages on the bonded brickwork were observed as compared to the CFRP-strengthened brickwork under cyclic compression loading. Additionally, the auxetic fabric application also increased the compressive resistance of brickwork by 38–60% under monotonic loading, which is comparably higher than with the CFRP strengthening technique.

## 1. Introduction

Unreinforced masonry (URM) has been widely used in heritage structures across the world as a construction material that needs continuous monitoring and intervention to withstand any unforeseen actions. The URM walls of these buildings, on many occasions, were constructed of bonded brickwork walls, consisting of two or more bricks layers in thickness. These bonded URM buildings usually suffer severe damage under seismic actions due to brittleness, poor connectivity/tensile strength and low energy dissipation capacity. Therefore, it is necessary to evaluate the optimum retrofitting techniques to minimise damage during seismic actions. Extensive research studies have been performed on the strengthening of URM walls to mitigate damages against seismic and other extreme actions. More than 50,000 material types have been used in retrofitting, strengthening, design and as an energy-absorbing protective layer on the surface of structures [1]. Carbon Fibre-Reinforced Polymer (CFRP) is a composite material which is a combination of carbon fibre and a polymer matrix (elastomers, thermoset and thermoplastic) or ceramic, and it is the most widely used strengthening material. Other commonly used fibrous composites are glass, basalt aramid and steel fibre composites, whereas common polymer matrices are esters and epoxies [2,3]. The choice of fibrous composite is based on cost-effectiveness without compromising mechanical properties, fatigue and corrosive resistance. In contrast, glass fibre-reinforced polymer is relatively cheap but lacks the interface bonding strength of the fibre/resin [3]. To resolve this, some researchers [4,5,6,7] have suggested a polymer that combines carbon and glass fibres prepared by pultrusion technology. The carbon/glass Hybrid Fibre-Reinforced Polymer (HFRP) can significantly reduce the amount of carbon fibre and reduce the material cost without compromising the effectiveness of the mechanical properties of the polymer. 

Carbon Fibre-Reinforced Polymers (CFRP) have been widely used to strengthen masonry structures under lateral loads due to their advantages of high strength, light weight, and corrosion resistance characteristics [8,9,10,11,12,13,14,15]. However, the use of CFRPs to improve the compression behaviour of bonded brickwork has been scarcely investigated in previous studies. Thamboo and Dhanasekar [16] tested bonded brickwork under cyclic compression and showed that various types of bonded brick masonry were vulnerable to cyclic actions. Later on, they employed carbon CFRP laminates to strengthen bonded brickwork wallettes under concentric and eccentric compression loads and reported no benefit due to early de-bonding of the fabric from the masonry [17]. The other efficient strengthening technique developed in recent years is a passive, externally bonded prestressed CFRP plates anchor to the structure using epoxy adhesive [18,19]. The main advantage of using this system is a reduction in premature debonding failures. as well as delaying the onset of material cracking. Further, Mahmood and Ingham [20] performed a comparative study for different sizes of CFRP strips on the substrate of the masonry wall under in-plane shear. The strip application in the horizontal, vertical, or diagonal directions directly influenced the in-plane strength of masonry walls. This research showed that CFRP application would enhance the in-plane strength, stiffness, and toughness of masonry walls. With the advantage of strength enhancement, the CFRP-retrofitted wall also suffered from fabric de-bonding. In general, high-adhesive epoxy-based compounds are used to apply CFRP on the surface of masonry walls. However, the use of adhesives on the surface of historic masonry buildings is not encouraged due to poor temperature resistance, low vapour permeability and poor chemical incompatibility with the masonry substrate. CFRP strips with adhesives introduce a deep impregnation, which makes the epoxy almost impossible to remove, and creates durability concerns due to the gluing system [21,22]. In addition, certain properties of clay masonry, such as porosity and surface unevenness or roughness, also affect the epoxy bonding between the fabric and the brick substrate, as well as restrictions related to intervention strategies for historic masonry structures [23]. Subsequently, in order to strengthen heritage masonry structures, cementitious-based materials such as fibre-reinforced cementitious matrix (FRCM) systems with open fabrics and cementitious matrices are being widely recommended in place of CFRP systems [24,25,26,27,28,29,30]. However, strengthening masonry samples with FRCMs produced a relatively lower strength gain compared to CFRP alternatives. The cement-based matrices lack the effectiveness of binding as compared to epoxies. This weakness is due to the result of mortar granularity, which hinders the penetration and impregnation of fibre sheets. Hence, FRCM strengthening is not an optimum solution in all the retrofitting scenarios. 

Therefore, the quality and type of adhesive/matrix plays a crucial role in the effectiveness of fabric-bonding with heritage structures. As a result, different types of epoxy polymers have been used as an adhesive bonding to preserve the architectural or aesthetic view of buildings [21]. Different polymeric fabrics with adhesive epoxy can increase the compressive, shear and bending strength without increasing the structural mass under cyclic/seismic, impact and blast loading [31,32,33,34,35,36]. However, due to large difference in the stiffness and strength characteristics of the masonry substrate and the CFRP laminates, their interface suffers significant shear stress and is susceptible to premature delamination. Asad et al. [37,38] have demonstrated such delamination due to widely different stiffnesses of the mortar matrix and the fabric under flexural testing. Delamination limits the composite action, leads to an abrupt fibre fracture and reduces the ductility and energy dissipation characteristics of the strengthened masonry. Considering these weaknesses, externally bonded CFRP fabric with adhesive epoxy is limited to only few applications. However, it is also vital to fill the gap of knowledge in the literature on the evaluation of failure modes of the wallettes under monotonic and cyclic compression loading using the finite-element method. Additionally, to eliminate such delamination, a novel material is required that can sustain seismic loading without de-bonding, can dissipate significant seismic energy, and simultaneously be lighter in weight. To achieve these objectives in the strengthening of masonry, the application of auxetic materials possessing a negative Poisson’s Ratio (NPR) on the masonry substrate can improve the performance of walls during cyclic/seismic loading. 

Auxetic fabrics with NPR experience elongation (or contraction) in both in-plane directions when they are stretched (or compressed). Different auxetic geometries with various modifications have been used in mitigating structural damage under seismic loading [39,40,41,42], for the crashworthiness of road vehicles [43], in aerospace engineering structures [44], in biomedical engineering [45,46,47] and in textile technology, indicating a wide range of applications [48]. These industries have prospered due to the enhancement of mechanical properties such as fracture, toughness, indentation resistance, shear modulus and vibration absorption [49,50,51]. To the best of the authors’ knowledge, only limited research on their application in civil engineering has taken place recently [37,38,52,53,54]. Asad et al. [37] demonstrated that under flexural loading, the energy dissipated by the auxetic fabric-overlayed mortar matrix composite specimen was about 70% higher compared to the CFRP fabric composite specimens. The higher energy absorption of the auxetic fabric composite was due to non-debonding of the interface at a high rate of deflection. Resistance to debonding enabled the auxetic fabric-rendered masonry walls to perform superiorly in contrast to the CFRP-rendered masonry wall in terms of out-of-plane deflection at different impact velocities [38]. Zahra et al. [52] discuss that the embedment of the auxetic fabric between the bed joints of dry stack blocks were beneficial in reducing peak contact pressure. The peak contact stress between the dry stack blocks was reduced by about 50% due to the insertion of the auxetic fabric. Additionally, Zahra et al. [53] compared mortar-auxetic composites with mortar-fibre glass composites under compression load. The NPR effect of the auxetic foam showed considerable alteration of the mortar composites’ behaviour by eliminating delamination at the interface. The current paper is an attempt to numerically examine the effect of the application of an auxetic fabric layup to the surface of masonry wallettes to mitigate damages under cyclic compression loading in comparison to CFRP strengthening. 

Subsequently, a research study was carried out to evaluate the performance of auxetic fabric layups on the surface of bonded brick masonry wallettes under cyclic compression in comparison to CFRP layups. In the literature, two modelling approaches are common for modelling masonry walls when strengthened with composite materials. The first approach is homogenous macro-modelling [55,56], which allows models to simulate the brick and mortar together as one homogenous material. In contrast, the second approach is micro-modelling of masonry walls, which presents the brick and mortar as individual components in the modelling [57,58]. Additionally, in [59,60,61,62,63], researchers have developed 3D finite-element models of nacre-like microstructures that have some random and statistical variations in the microstructure of staggered composites to accurately capture the fracture and fine-crack propagation in the individual brick or mortar. These models were developed in multiple stages, and the first stage is the development of a small RVE (Representative Volume Element) to capture the effects of defect and localisation. The second stage is where large-scale, discrete element models of staggered composites with statistical variation are applied to capture the failure of composites. Considering all these modelling techniques, the homogenous macro-modelling approach is usually adapted for large-scale structure modelling when the focus of the study is to evaluate only the global behaviour [38]. The micro-modelling approach, although computationally expensive, can be utilised for small-scale studies to understand local failure behaviours such as debonding or delamination [57]. As the main aim of this paper is to predict the effectiveness of CFRP fabric and auxetic fabric application on masonry wallettes under compression loading, the macro-modelling technique provided a simpler model with reasonable failure characterisation to represent the delamination of strengthening fabrics from the masonry wallettes. 

Thus, in this study, a simplified finite element (FE)-based micro-modelling technique was developed to analyse the bonded brickwork with and without strengthening materials subjected to monotonic and cyclic compression loads. The formulated FE modelling technique of the bonded brickwork was verified with the experimental results published earlier by the authors [16]. Later, the validated numerical models of the masonry wallettes were expanded with a single layer of CFRP laminates on the surfaces of the wallettes. The model of CFRP-strengthened masonry developed was validated with the experimental results obtained by the authors [17]. Finally, the validated CFRP strengthening layer on the brickwork masonry wallette was replaced with auxetic fabric layers, and then the compression resistance and mechanical characteristics of the wallettes under both monotonic and cyclic compression loadings were examined. Thus, this study contributes to the knowledge on the effectiveness of these two externally bonded strengthening techniques (i.e., CFRP layup and auxetic fabric layup) on masonry under axial compression.

## 2. Numerical Modelling of URM Axial Compression

A simplified 3D micro-mechanical numerical modelling technique was adopted in this paper to analyse double brick-bonded masonry wallettes under both monotonic and cyclic compression. An implicit finite-element modelling method was developed to simulate the response of masonry wallettes under axial monotonic and cyclic displacement. The FE model incorporated the material and contact non-linearities, and the input parameters were calibrated using the datasets of the wallette properties obtained through experimental study [16]. Figure 1 shows the FE model developed for the double-brick wallettes constructed of two types of brick units (namely B1 and B2, described in the following sub-section).

The mortar joints were modelled as zero-thickness interfaces with non-linear damaging properties, while the bricks were enlarged to incorporate half of the thickness of mortar joints all around the bricks, as shown in Figure 1. This type of modelling has been adopted earlier to model bonded brickwork under monotonic compression [64]. The loading plates as illustrated in Figure 1 could move in the direction of compression loading and unloading at the rate of 0.25 mm/min. The experimental programme, model geometry, constitutive material models and validation results are discussed in detail in the following sub-sections.

### 2.1. Experimental Programme

Experimental results for masonry under monotonic and cyclic compression established by the authors [16] were used to validate the applicability of the established numerical modelling technique. The masonry wallettes in this experimental programme were constructed using two types of clay brick units, i.e., B1 and B2 bricks. The difference in the mechanical properties of the B1 and B2 bricks lay mainly in the manufacturing process, i.e., controlled and uncontrolled firing in the kiln. However, the B1 bricks with low strength and modulus of elasticity are very common in many developing countries. In total, 14 masonry wallettes were constructed using B1 and B2 clay bricks with double brickwork thickness to assess the monotonic and cyclic behaviour of bonded masonry. The wallettes were built and tested according to BS EN 1052-1 [65] and loaded under the monotonic and cyclic displacement-controlled protocols. The thickness of the mortar was strictly maintained at 10 mm with the help of an experienced mason. The dimensions and the average compressive strengths of the wallettes are presented in Table 1 along with their coefficients of variation (COV). The experimental representation with the numerical model of the double layer-bonded brickworks constructed of B1 and B2 brick types are shown in Figure 2.

The wallettes were tested using a 1000 kN-capacity servo-controlled Universal Testing Machine (UTM) to assess the monotonic and cyclic compressive strengths and deformation characteristics. Five-millimetre-thick plywood was used between the top and bottom platens of the UTM to uniformly distribute the compression loading over the wallettes. In total, eight displacement transducers were attached to record the axial and lateral deformation in each wallette. Four transducers were attached vertically on both faces (two on each face) of the wallette to record the axial deformation, as shown in Figure 2a,b. Similarly, four remaining transducers (two on each face) were placed horizontally to record lateral deformation on each face of the wallette. A uniform monotonic displacement of 10 mm was applied on the top of the loading plate, with boundary constraints marked with a reference node to simulate the compression tests, as shown in Figure 2. For cyclic testing, a typical cyclic loading protocol in terms of displacement was applied using the same reference node as shown in Figure 3. The cyclic displacement procedure was taken from the load-displacement responses of similar wallettes tested under monotonic loads, as presented in previously published articles [16,17,64]. The elastic, hardening and peak limits were identified from the load-displacement response. The elastic limit point was observed at one-third of the peak load, and the hardening limit was considered as 0.8 times the peak load [16,17,64]. The considered number of cycles within each limit are mentioned in Table 2. 

### 2.2. Model Geometry Details

For the simplified 3D micro-mechanical numerical modelling method, the homogenised brick units B1 and B2 were modelled using 3D solid elements (C3D8R) with 8 nodes and 24 degrees of freedom. The unit size was represented by an enlarged brick consisting of full-scale brick enveloped by a half-thickness of the mortar bedding layer, as shown in Figure 2. These masonry bricks were arranged in multiple layers using zero-thickness cohesive interface elements to simulate the bond behaviour under shear, tension, and compression actions. A mesh convergence study was performed for brick units (B1 and B2) to obtain accurate and computationally efficient results. Subsequently, a mesh size of 20 mm × 20 mm × 20 mm was employed for the brick elements after the mesh convergence studies had been carried out.

### 2.3. Material Properties and Constitutive Model

A plasticity-based damage constitutive model to represent the damage in the masonry under cyclic compression loading was employed. The concrete damage plasticity (CDP) model in-built in ABAQUS FE software [66] was used to simulate the mechanical behaviour of smeared brick and mortar together. This model can predict two main failure modes: tensile cracking and compression crushing. Beyond the failure limit, the formation of minor and major cracks was represented with a softening stress-strain response, as shown in Figure 4. After calibrating with the experimental results, a threshold strain level of 0.05 was used to enact the element deletion technique for initiating the brittle crack opening in the masonry units. The element deletion limit value was defined from the studies performed in the past by the authors [36,66,67,68]. The constitutive model used for the B1 and B2 brick wallettes is shown in Figure 4. The elastic modulus of the B1 series wallette was input as 200 MPa, while for the B2 series wallette, an average value of 995 MPa was adopted from the authors’ previous experimental study [16]. The Poisson’s ratio was kept as 0.18, as was observed from the experiments. In addition, the CDP model depends on five main parameters: dilation angle (*ψ*) to account for the volume change caused by the inelastic behaviour of bricks; flow potential eccentricity (*e*), which defines the rate at which the hyperbolic flow potential approaches its asymptote; the ratio of the biaxial compressive strength of the material to its uniaxial compressive strength ratio (*f_b_*_0_*/f_c_*_0_); shape factor (*K_c_*), which is the ratio of the tensile to the compressive meridian for describing the shape of the yield surface, and viscosity (*µ*), which represents the relaxation time of the visco-plastic system and is used for the visco-plastic regularisation. More details of these parameters can be found in [69,70]. The assigned CDP parameters are listed in Table 3. 

### 2.4. Zero Thickness Interface: Cohesive Zone Model

The zero-thickness interface between the homogenised units was simulated using a cohesive zone model. The interaction was modelled using the simple bi-linear separation constitutive law as shown in Figure 5. The traction law is graphically represented as the traction force (*τ*) resulting in the opening displacement (*δ*). The interface allowed damage initiation followed by damage evolution until the total degradation of surface bonding. The elastic behaviour of the interface relates the normal and the shear (tractions) to the normal and shear displacements (separation) across the interface.

The normal tensile $\left(K_{nn}\right)$ and shear $\left(K_{ss},\;K_{ss}\right)$ stiffness of the interface are related to the properties of mortar used for connecting the bricks. The lime mortar adopted in this research has low tensile strength and slightly higher shear stiffness. The input values for the interface modelling are provided in Table 4. The damage initiation in the interface was assumed to occur when quadratic traction factors involving the combined normal stress $\left(\sigma_{n}\right)$ and shear stress $\left(\tau_{s},\;\tau_{t}\right)$ ratios reached the value 1 as shown in Equation (1)
(1)$${\left\{\frac{\langle \sigma_{n}\rangle }{\sigma_{n}^{0}}\right\}}^{2}+{\left\{\frac{\langle \tau_{s}\rangle }{\tau_{s}^{0}}\right\}}^{2}+{\left\{\frac{\langle \tau_{t}\rangle }{\tau_{t}^{0}}\right\}}^{2}=1$$
where $\sigma_{n}^{0}$ is the limiting (maximum) normal strength and $\tau_{s}^{0},\tau_{t}^{0}$ are the limiting (maximum) shear strength of the interface mortar. The interface damage evolution is expressed in terms of fracture energy release as shown in Equation (2)
(2)$$G_{n}^{c}+(G_{s}^{c}-G_{t}^{c}){\left(\frac{G_{s}+G_{t}}{G_{n}+G_{s}+G_{t}}\right)}^{\eta }=G^{c}$$
where $G_{n}$, $G_{s}$ and $G_{t}$ are work done in normal and shear directions while $G_{t}^{c}$ and $G_{s}^{c}$ are the corresponding maximum fracture energies, which cause failures in normal and shear directions, respectively. The dependency of the fracture energy η is a material parameter proposed by Benzeggagah & Kenane [71] and was taken as 1.5. The Benzeggagah–Kenane [71] fracture criterion defines the critical fracture energies when the deformation and the first and second shear directions are the same. The properties of the zero-thickness interface to account for cohesion, traction separation law and friction are listed in Table 4. Since the masonry assemblages considered to validate the numerical models were made of lime-mortared clay bricks with low tensile strength, the adopted interfacial properties are relatively lower than for the conventional masonry made with cement mortars [70,71,72].

### 2.5. Validation Results

This section presents the validation of the FE model developed for the double-brick masonry wallettes under monotonic and cyclic compression loading using the experimental datasets presented in [16]. The validation includes the failure modes of bonded brickwork wallettes, stress-strain curve validation with the experimental results and the strain variation in the region of significant damage.

#### 2.5.1. Failure Modes

The validation of the FE results under monotonic loading is discussed in this section. However, the detailed study of the monotonic testing results, including the effect of geometry and thickness of wallettes, have been presented in the authors’ previous research [64]. The failure modes of the double-thick brickwork models of the B1 and B2 series are presented in terms of maximum principal strain in Figure 6. The numerically predicted failure patterns of the double-thick brick masonry wallettes are shown to be in close agreement with the experimental failure modes (as shown in Figure 6a,b). The FE result also predicted similar vertical cracking on the front face and tensile splitting on the side faces. The tensile cracking phenomena of masonry under axial compression are well understood to be due to incompatible deformation characteristics between unit and mortar and subsequent lateral tensile strains caused by a Poisson’s ratio effect which induces vertical tensile splitting cracks [73,74]. The maximum principal strain contour plot of magnitude 0.009 shows that the masonry wallettes of brick type B2 (Figure 6b) were more brittle than the B1 brick wallettes. These failure strain limit values can also be correlated with the axial stress-strain plot of the wallettes under the monotonic compression load shown in below Section 2.5.2.

The failure modes of double-thick bonded brickwork models of the B1 and B2 series under cyclic compression are shown in Figure 7 also in terms of principal strain. The failure patterns observed in the numerical model for both wallettes are shown to be in close agreement with the experimental failure modes. The wallettes under cyclic loading observed similar single splitting cracks through the thickness of the wallette, and parallel tensile cracks were observed along the width of the wallettes. The vertical cracks were mainly observed in the central area and near the edges of the faces. After the peak load, these cracks propagated diagonally and opened, leading to spalling from the edge of the wallettes. The post-peak stage under cyclic loading resulted in wider crack propagation within a short period, associated with the spalling of brick-and-mortar pieces.

#### 2.5.2. Monotonic and Cyclic Stress-Strain Relationship

The axial compressive stress vs strain curves obtained from the numerical models and experimental data are presented in Figure 8 for B1 and B2 brick wallettes. To plot the numerical stress-strain curves, the axial (vertical) deflection from the nodes equivalent to the gauge length used in the experiments [16] was used to calculate the axial strain, while the reaction force was used to find the axial stress by dividing it with the cross-section area of the respective specimens. The lateral strains represent the horizontal strain measurement on the front face of the wallette samples, as indicated in Figure 2. The predicted stress-strain curves observed from the numerical models match reasonably well with the experimental curves for both types of brick, B1 and B2.

Figure 9 shows the cyclic stress-strain curves of the bonded brickwork of the B1 and B2 series. The axial stress-strain curves derived from the numerical models were also compared with the experimental curves for both brickwork specimens. From the cyclic envelope stress-strain curves, the elastic moduli were derived, which varied from 135 MPa for B1 series wallettes to 506 MPa for B2 series wallettes. It can be clearly seen that the non-linear behaviour under cyclically loaded wallettes started nearly around 40% to 50% of peak compressive stress, which was associated with the initiation of cracks in the wallettes. The accumulation of non-reversible axial strains in the masonry wallettes resulted in the degradation of stiffness in each cycle, especially after about 40% of peak stress, for both B1 and B2 series wallettes. This degradation indicates the gradual damage that has occurred in each cycle in the masonry wallettes. Therefore, the pattern of cyclic stress-strain curves of wallettes shown in Figure 9 justifies the overall decrease in the compressive strength of masonry wallettes under cyclic compression when compared to monotonic loading [16]. 

Additionally, Figure 9 shows that the B2 series wallettes have significantly lower axial and lateral deformability than the B1 series wallettes. This phenomenon is due to high strength and lower deformability properties of B2 bricks, which influenced the overall stress-strain behaviour of the masonry. Hence, the accumulation of strain and the time history of the strains is significant for the compressive strength of the masonry wallettes, as well as for defining mitigating strategies under cyclic loading. 

## 3. Numerical Modelling of Strengthened Brick Wallettes under Axial Compression

This section entails the mitigating methods to enhance the performance of brick masonry under axial compression loading. The technique involves a layup of CFRP laminates on the faces of the bonded brickwork wallettes, as shown in Figure 10, and its effectiveness was studied using numerical modelling. The fabric material properties and the adhesion properties of the epoxy resin are discussed. This section also describes the adhesive interface modelling between the surface of masonry and CFRP-composite rendering under monotonic and cyclic compression testing. Moreover, the validation of the modelling technique for CFRP layup with the experimental results under monotonic loading, and then predicting the behaviour of CFRP layup for the same kind of masonry wallettes under cyclic compression testing, are presented in this section.

### 3.1. CFRP Fabric: Geometric Detailing

The fabric geometry was modelled using the conventional planar 3D shell element (S4R) with six DOF at each node [38,54,75,76], as shown in Figure 10. The fabric was assigned a homogenous laminar property at each layer using five integration points through the thickness. A single layer of fabric layup on the front and rear face of the masonry wallettes was employed. The dimension of the CFRP fabric used for bonded bricks wallettes of the B1 and B2 series were 410 mm (L)×730 mm (H)×1 mm (T) and 430 mm (L)×750 mm (H)×1 mm (T), respectively, as shown in Figure 10.

### 3.2. CFRP Fabric: Material Properties

The Hashin’s damage model available in ABAQUS [66] was used to simulate the characteristics of the CFRP fabric [77]. Hashin’s damage initiation and evolution laws were used to define the progressive damage of CFRP fabrics. Equations (3)–(6) were used to define the damage initiation and the damage evolution, which were adapted from the study by Asad et al. [38]. The maximum degradation was considered for the fabric when the damage variable *D* reaches a magnitude of 1. The element deletion technique was only activated when the damage variable achieved this value. A failure is initiated in the fabric laminate when one of the following indices is greater than or equal to 1: fibre tension, fibre compression, matrix tension and matrix compression, as represented by Equations (3)–(6), respectively.
(3)$$\mathrm{Fibre}\;\mathrm{tension}\;\mathrm{failure}\;(\sigma_{xx}\geq 0).\;F_{Fiber\;Tension}={\left(\frac{\sigma_{xx}}{X^{T}}\right)}^{2}+\alpha_{h}{\left(\frac{\tau_{xy}}{S^{L}}\right)}^{2}$$
(4)$$\mathrm{Fibre}\;\mathrm{compression}\;\mathrm{failure}\;(\sigma_{xx}<0)\;F_{Fiber\;Compression}={\left(\frac{\sigma_{xx}}{X^{c}}\right)}^{2}$$
(5)$$\mathrm{Matrix}\;\mathrm{Tension}\;(\sigma_{yy}\geq 0)\;F_{Matrix\;Tension}={\left(\frac{\sigma_{yy}}{Y^{T}}\right)}^{2}+{\left(\frac{\tau_{xy}}{S^{L}}\right)}^{2}$$
(6)$$\mathrm{Matrix}\;\mathrm{Compression}\;(\sigma_{yy}<0)\;F_{Matrix\;Compression}={\left(\frac{\sigma_{yy}}{2S^{T}}\right)}^{2}+\left({\left(\frac{Y^{C}}{2S^{T}}\right)}^{2}-1\right)\frac{\sigma_{yy}}{Y^{C}}+{\left(\frac{\tau_{xy}}{S^{L}}\right)}^{2}$$
where $X^{T}$ denotes the tensile strength in the fibre direction, $X^{c}$ denotes the compressive strength in the fibre direction (which is assumed to be negligible in the analyses), $Y^{T}$ and $Y^{C}$ are the tensile and compressive strengths in the direction perpendicular to the fibres ($Y^{C}$ also assumed negligible in the analyses), $S^{T}$ denotes the transverse shear strength, $S^{L}$ denotes the longitudinal shear strength and $\alpha_{h}$ is the coefficient that determines the contribution of shear stress to the fibre tensile failure initiation criteria. Furthermore, $\sigma_{xx}$, $\sigma_{yy}$ & $\tau_{xy}$ are the components of the effective stress tensor. The input properties of the CFRP laminate were taken from past experimental studies [73,74,78,79]. The input properties of the CFRP fabric are listed in Table 5, Table 6 and Table 7. It has to be mentioned that the compressive strength of CFRP fabric is not explicitly considered for design or analyses. Although FRP fabrics are compressed in the strengthened masonry, the FRP failures were mainly dominated by tensile rupture or delamination due to the dilation of the strengthened assembly under axial compression. 

### 3.3. CFRP Fabric Brick Wallettes: Interface Modelling

From the literature [17,37,38], delamination/de-bonding was observed between CFRP surfaces and the masonry wallettes. Such delamination or de-bonding at the interfaces was mainly due to the adhesive failure between the epoxy-coated CFRP and masonry substrates. In numerical models, the de-bonding was simulated using the cohesive zone interface model. The interaction features in the cohesive model were evaluated using the same bi-linear traction separation constitutive law used for zero-thickness mortar bedding, as shown in Figure 5b. The cohesive properties between the CFRP fabric and the masonry wallettes are summarised in Table 8.

### 3.4. Validation of the CFRP Fabric-Strengthened Brick Wallettes under Monotonic Compression Loading

The developed numerical model for CFRP-strengthened brickwork wallettes was validated with the experimental results presented in Thamboo et al. [17]. The experimental programme involved testing of six CFRP-strengthened masonry wallettes under monotonic axial static loading. The masonry wallettes considered for the testing were made from the same types of bricks (B1 and B2 clay brick units) as discussed in Section 2. Hence, the constitutive material model was duplicated, as explained in detail in Section 2.3 for the masonry wallettes under monotonic and cyclic loading. However, the application of the CFRP and the testing procedures were explained in detail in Thamboo et al. [17]. The unidirectional CFRP sheet layups were applied using saturated epoxy resin as an adhesive layer on the front and the back surface of the masonry wallettes, as shown in Figure 11. It must be mentioned that the unidirectional CFRP sheets were applied parallel to the bed joints, a technique which is commonly employed to improve the shear/tensile resistance of masonry walls. Additionally, a confining of walls with fabrics as is usually carried out for column elements is not possible due to their considerably larger length than columns and connections with crossing walls. The panels were then tested under axial compression to verify the influence of the conventional strengthening method (with unidirectional CFRP parallel to bed joints) on the compression strength characteristics of masonry, which is a prime input parameter for the macro-modelling of masonry structures.

The testing was carried out at a constant displacement rate of 0.25 mm/min using the same UTM of 1000 kN capacity to capture the complete post-crack behaviour of masonry wallettes under axial compression. The numerical model developed corresponding to the experimental test setup is shown in Figure 10. Displacement transducers were attached to the wallettes to capture the vertical deformation at mid-height of the wallettes, as shown in Figure 11c. The typical failure modes of monotonically loaded wallettes of B1 and B2 brick masonry strengthened with CFRP sheets are shown in Figure 12. Due to the CFRP sheets attached to the front and back face of the masonry wallettes, no visible cracks were noted in the experimental test, as shown in Figure 12. Previous similar research studies on multiple thin CFRP strips layups on masonry walls [80,81] were verified to confirm the failure modes developed on the side and front/rear face of the wallettes. The failure modes on the sides with visible CFRP deformation buckling are shown in Figure 13a. A premature interaction failure at the side edge between the CFRP strip and the masonry surface is shown in Figure 13b. 

The failure modes obtained from the FE model of both types of brick (B1 and B2) masonry wallettes strengthened with CFRP fabric are shown in Figure 14. The strain contour plots show that the maximum stress concentration occurred at the vertical edge of the wall, which resulted in the vertical cracks due to splitting. The magnitude of the strain observed in the B2 brick-type masonry wallettes was smaller than in the B1-brick type masonry wallettes, due to premature failure observed at the interface. This was mainly due to the brittle nature of the B2 bricks, which failed at a lower strain, compared to the B1 brick-type masonry wallettes. 

The numerically predicted stress-strain curves were also compared with the experimental curves for the CFRP-strengthened masonry wallettes, as shown in Figure 15. For plotting the numerical stress-strain curves, the axial (vertical) deflection from the nodes for the same gauge length that was used in the experiments [17] was used to calculate the axial strain, while the reaction force on the loading plate was used to find the axial stress by dividing it with the cross-section area of the respective specimens. The lateral strains shown in Figure 15 represent horizontal strain measured on the front face of the CFRP sheet samples. The same gauge length as in the experiments was used to calculate the horizontal lateral strains from the models. The numerical model results matched the experimental curves reasonably well, despite the variation in brick strengths. The stress-strain curve of the CFRP-strengthened wallette of B2 bricks, as shown in Figure 15b, confirmed the sudden or premature failure in these wallettes due to the brittleness of the B2 brick types.

### 3.5. Prediction of Cyclic Behaviour of CFRP Fabric-Strengthened Brickwork Wallettes

The numerical models were modified with the cyclic loading protocol (as shown in Figure 3) to study the effectiveness of strengthening under cyclic compression. This sub-section presents the performance of the CFRP fabric-strengthened masonry wallettes under cyclic compression loading. The failure modes of the CFRP fabric-strengthened masonry wallettes constructed of B1 and B2 brick types are shown in Figure 16. The failure modes are presented in terms of maximum principal strain, and the images shown in Figure 16 are at the ultimate failure stage of the CFRP-strengthened masonry wallettes. Debonding in the CFRP-strengthened masonry wallettes of B2 type brickwork was observed at a magnitude 83% lower than the B1 type brickwork-constructed masonry wallettes. This reduction in strain in B2 bricks wallettes has already been discussed in the previous section under monotonic static loading. The CFRP-strengthened masonry wallettes of B2 bricks showed a 67% decrease in strain compared to B1 brick wallettes strengthened with CFRP fabric. Based on these results, it can be said that the formulated modelling technique appropriately predicted the failure pattern of the CFRP-strengthened masonry wallettes using different types of brick units.

The stress-strain curves of the CFRP-strengthened cyclically loaded wallettes are presented in Figure 17. The cyclic hysteresis stress-strain curves predicted that the CFRP-strengthened masonry wallettes behaved much more stiffly than the control (un-strengthened) masonry under cyclic loads. However, the CFRP-strengthened masonry wallettes of B2 brick showed a premature failure due to the brittle and stiff properties of the B2 bricks, as shown in Figure 17b. The reason could be the early de-bonding of the fabric from the surface of the masonry wallettes. This phenomenon was further investigated for both types of brick wallettes by studying the strain levels at the interface of fabric and masonry wallette. 

A simplified illustration of the axial strain distribution with respect to time on the surfaces of the masonry wallettes and the CFRP sheet under cyclic loading is plotted in Figure 18. The axial strain obtained from the numerical model was plotted at mid-height of the wallettes. The difference in strain levels of masonry and CFRP sheet in Figure 18a,b is clear evidence of debonding at the interface between the CFRP fabric and the masonry wallettes. It is also proven from Figure 18 that the damage in B2 brick wallettes occurred at a much lower strain compared to the B1 bricks wallettes, due to the brittle nature of the B2 bricks.

## 4. Alternative Strengthening of Masonry Using Auxetic Fabrics

From the results of Section 3, it can be easily concluded that the CFRP strengthening of the masonry wallettes was not fully effective due to the early debonding of the CFRP fabric from the surface of the masonry wallettes under monotonic and cyclic compression loading. Alternative to these failure modes, it was observed from the literature [37,54] that an innovative auxetic fabric performed reasonably well when compared with CFRPs due to better bonding properties of the auxetic fabric. The same auxetic fabric was employed here to evaluate the behaviour of the auxetic fabric layup on the surface of the brick masonry wallettes under monotonic and cyclic loading.

### 4.1. Auxetic Fabric-Strengthened Masonry Wallettes under Compression Loading

The numerical model of auxetic fabric-strengthened masonry wallettes was developed by replacing the CFRP fabric with the auxetic fabric, which was also modelled using Hashin’s damage criteria [77] inbuilt in ABAQUS [66]. The properties of auxetic fabric are presented in Table 9, Table 10 and Table 11, which were taken from Asad et al. [37,38,76], and were validated earlier in [38] with experimental results [52]. It can be mentioned that for the masonry applications under axial compression, the fabrics mostly fail by tensile rupture or delamination rather than in compression, due to dilation of the strengthened masonry. Furthermore, the interface modelling and cohesive properties were assumed to be the same as used to validate the CFRP-strengthened masonry wallettes for both types of brick units, due to the same epoxy used for the bonding.

The failure modes of the auxetic fabric-strengthened masonry wallettes under monotonic loading for both types of bricks are shown in Figure 19 in terms of maximum principal strain. The predicted numerical failure patterns on the surface of the auxetic fabric, as shown in Figure 19a, imply effectiveness over the CFRP-strengthened masonry wallettes (Figure 14) under monotonic loading. The uniform distribution of the principal strains on the surface of the auxetic fabric compared to the CFRP-strengthened masonry wallettes increased the ductility of the wallettes. The maximum increase in the ductility (corresponding to an increase in ultimate failure strain) was, for the B1 brickwork wallettes, due to the non-debonding of the fabric. However, compared to the B1 brickwork wallettes, the B2 brick wallettes, as shown in Figure 19b, exhibited lower ductility due to premature debonding of the auxetic fabric. However, the results show that B2 brick assemblies were still more efficient when compared to the CFRP-strengthened B2 brick wallettes.

The numerically derived axial stress-strain curves of auxetic-strengthened wallettes are shown in Figure 20, and are compared to the CFRP-strengthened wallette results. Figure 20a shows an enhancement in strength with a significant increase in strain for the auxetic-strengthened masonry wallettes constructed of B1 type bricks. The enhancement is mainly due to the non-debonding behaviour of auxetic fabric on the surface of masonry wallettes. In comparison, the B2 type auxetic-strengthened masonry wallettes exhibited a slight increase in strength with a 67% increment in strain compared to the CFRP-strengthened masonry wallettes. Again, the reason for the slight increment in strength could lie in the stiffness properties of the B2 bricks. However, the bonding between the auxetic fabric and B2 brick masonry wallette surfaces increased due to the negative Poisson’s ratio of the fabric.

### 4.2. Auxetic Fabric-Strengthened Masonry Wallettes under Cyclic Compression

The auxetic fabric-strengthened masonry wallette models were also analysed under the cyclic loading protocol (as shown in Figure 3) to study the effectiveness of auxetic strengthening under cyclic compression. The failure modes of the auxetic fabric-strengthened masonry wallettes constructed of B1 and B2 brick types are shown in Figure 21. The failure modes are presented in terms of maximum principal strain, and the images shown in Figure 21 are at the ultimate failure stage. Figure 21a shows the initiation of the debonding of the auxetic-strengthened masonry wallettes at the maximum principal strain of 0.055, which is 67% larger than with the CFRP sheet (Figure 16a). Additionally, a reduction of 60% in the principal strain was observed in the B2-type wallettes when compared with the B1-type brickwork masonry wallettes (as shown in Figure 21a,b). 

The stress-strain curves of the cyclically loaded auxetic fabric-strengthened wallettes are presented in Figure 22. The cyclic hysteresis stress-strain curves predicted that the auxetic-strengthened masonry wallettes behaved in a softer and more ductile way than the control and CFRP-strengthened masonry wallettes (presented in Figure 17). Additionally, as expected, the auxetic-strengthened masonry wallettes of B2 brick showed an early failure compared to the B1 brick wallettes, due to the brittle and stiff nature of the brick, as shown in Figure 22b. However, compared to CFRP-strengthened brickwork, the auxetic-strengthened masonry wallettes showed a significant improvement in terms of ductility. It can be noted that there is an increase of 50% in compression strains from un-strengthened to auxetic-strengthened wallettes of the B1 brick type, as shown in Figure 22a. In contrast, when compared with the CFRP-strengthened wallettes (presented in Figure 17), the same compression strain showed a significant increase of about 80%. A similar observation of enhancement in compression strain was observed for the auxetic-strengthened masonry wallettes of B2 brick types when compared to the corresponding un-strengthened masonry wallettes. Hence, to confirm the non-debonding characteristics of the auxetic fabric, an investigation was performed.

A simplified illustration of the axial strain distribution with respect to time on the surfaces of the masonry wallettes and the auxetic fabric sheet is plotted in Figure 23. The axial strain was plotted at mid-height of the wallettes. Figure 23a,b show that the strain distribution on the surface of the auxetic fabric and at the masonry surface completely overlapped with each other, with variation in time. The plotted strain distribution time history provides clear evidence of non-debonding at the interface between the auxetic fabric and the masonry wallettes. However, comparing Figure 23a,b, the damage in the auxetic-strengthened masonry wallettes constructed of B2 brick occurred at lower strain (0.005) compared to the B1 bricks wallettes (strain = 0.02) due to the stiff and brittle nature of the B2 bricks.

## 5. Summary and Conclusions

A simplified micro-modelling method of un-strengthened and CFRP-/auxetic fabric- strengthened masonry was developed for predicting the behaviour of bonded brickwork wallettes under monotonic and cyclic compression loading. The element deletion technique was applied for the 3D solid and shell elements to allow for damage in the brick units and the fabric sheets, respectively. The material and contact/interface nonlinearities were incorporated into the simplified micro-model developed. The modelling methodologies were validated using different experimental results of control/un-strengthened and CFRP-strengthened masonry wallettes under cyclic and monotonic loading. The validated models were used to systematically predict the monotonic and cyclic compression behaviour of auxetic fabric-strengthened masonry under monotonic and cyclic compression. Following are the main conclusions drawn from this research:▪The simplified micro-modelling technique for the masonry wallettes constructed with different brick types was found to be computationally efficient without compromising the accuracy in predicting compression behaviour under the monotonic and cyclic compression loading protocols. ▪Due to the de-bonding nature of failure in CFRP-strengthened masonry wallettes, the increases in the compressive resistance and ductility of masonry wallettes were limited. This phenomenon was more pronounced in the cyclic loading conditions, where with progressive damage, premature compression failures were observed due to early de-bonding.▪The CFRP strengthening increased the compressive resistance of the masonry wallettes compared to the un-strengthened masonry wallettes by about 10–20%. In contrast, the auxetic fabric application increased the compressive resistance of brickwork by 38–60% under monotonic loading.▪In cyclic compression loading, the auxetic fabric application to the surface of the masonry wallettes increased the ultimate strains without compromising the strength, compared to the un-strengthened masonry wallettes, which significantly enhanced the ductility (up to ~50%) of the masonry wallettes. 

From the evaluated results of this study, it can be concluded that the application of the auxetic fabric on the surface of the masonry wallettes is more efficient than conventional CFRP fabric in enhancing compressive resistance and ductility under monotonic and cyclic compression loading. This enhancement was achieved due to the negative Poisson’s ratio effect of the auxetic fabric, which ultimately improves the bonding nature. However, more experimental and numerical studies are needed to confirm the parametric analyses and to further experimentally verify the effectiveness of auxetic fabric application in strengthening masonry under different stress states in order to comprehend the complete characteristics. 

## Figures and Tables

**Figure 1 polymers-14-01800-f001:**
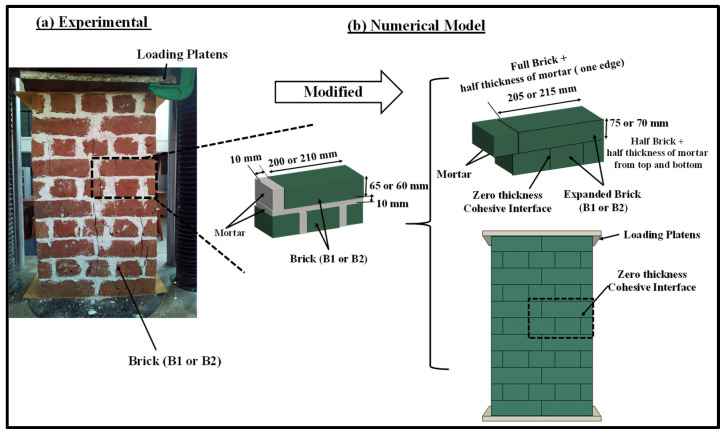
Numerical modelling of double-brick thick wallettes using expanded size of brick units and zero-thickness interfaces.

**Figure 2 polymers-14-01800-f002:**
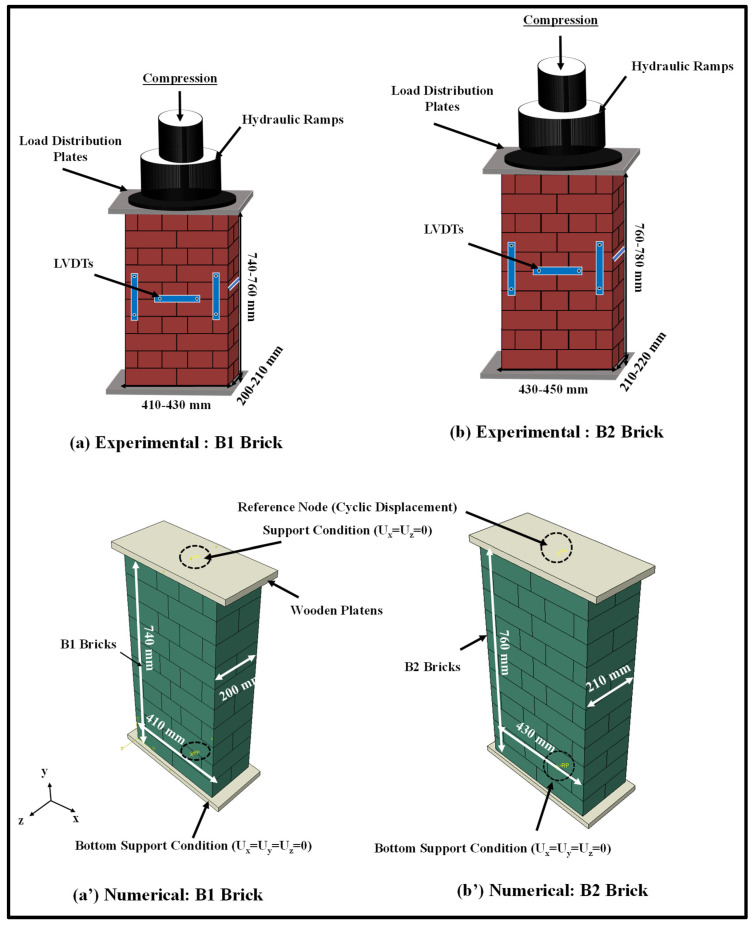
Geometrical representation of masonry wallettes during experimental setup and corresponding numerical representation; (**a**) B1 and (**b**) B2 wallettes.

**Figure 3 polymers-14-01800-f003:**
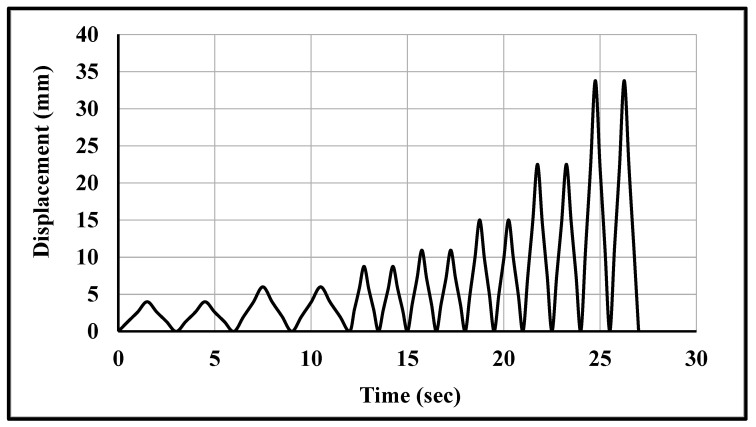
Cyclic loading protocol applied to the wallettes.

**Figure 4 polymers-14-01800-f004:**
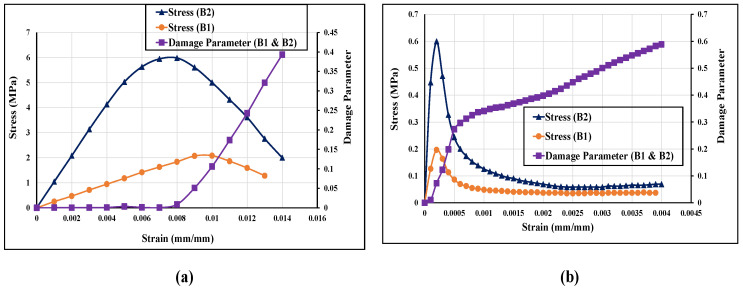
Input stress-strain and damage properties of brick wallettes; (**a**) compression and (**b**) tension.

**Figure 5 polymers-14-01800-f005:**
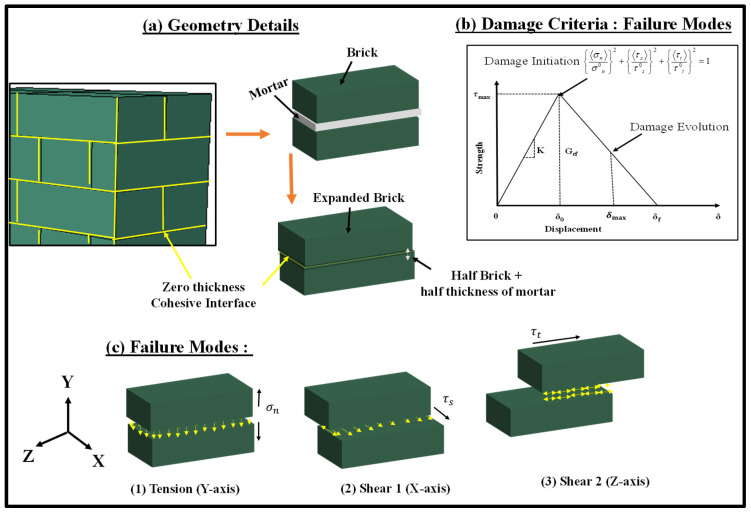
Zero-thickness Cohesive Interface model details; (**a**) geometry details, (**b**) damage criteria at interface and (**c**) failure modes at the interface.

**Figure 6 polymers-14-01800-f006:**
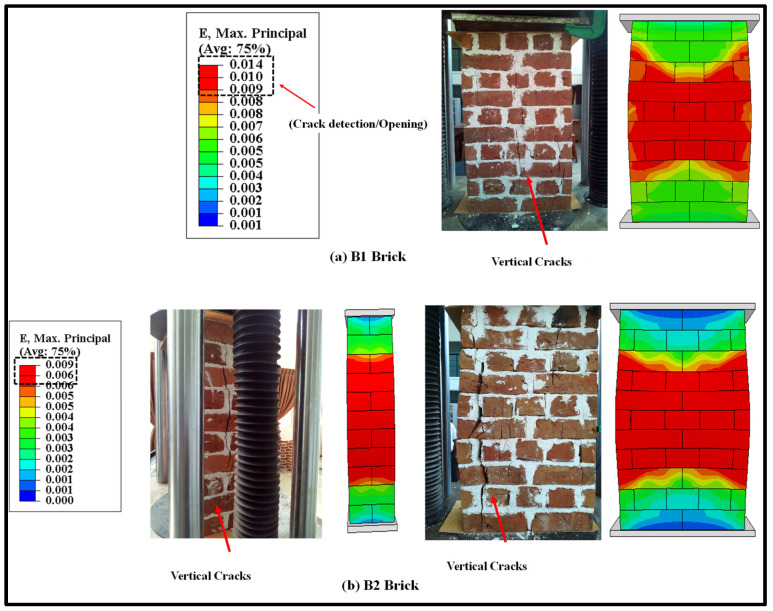
Predicted failure modes of the wallettes compared to the experimental results under monotonic compression test; (**a**) B1 and (**b**) B2 wallettes.

**Figure 7 polymers-14-01800-f007:**
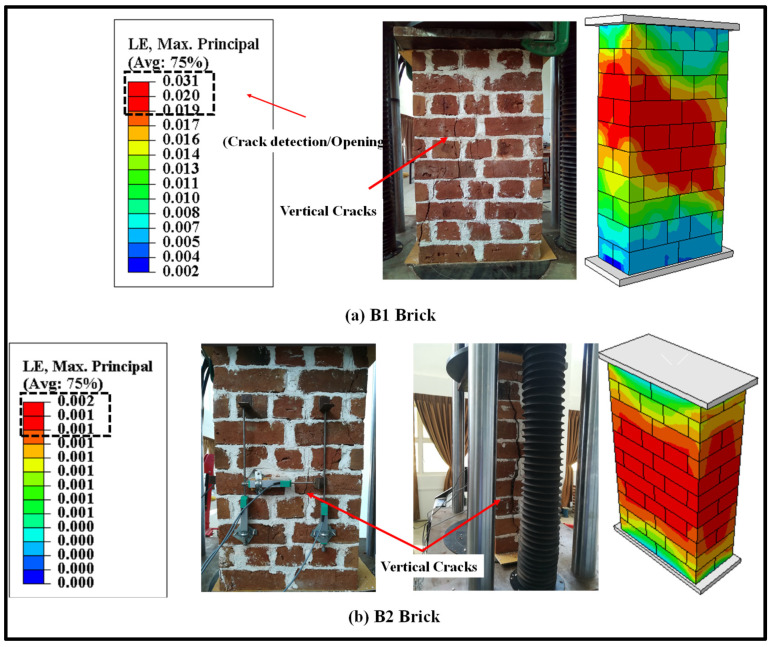
Predicted failure modes of the wallettes compared to the experimental results under cyclic compression test; (**a**) B1 and (**b**) B2 wallettes.

**Figure 8 polymers-14-01800-f008:**
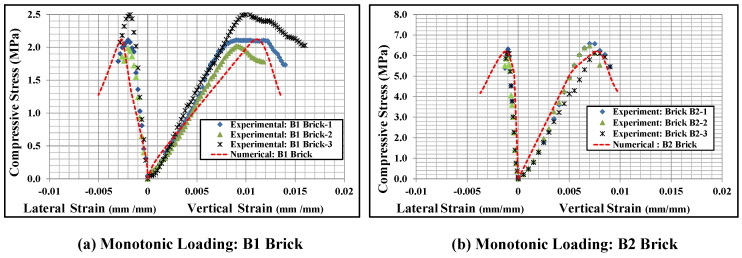
Stress-strain responses of the wallettes under monotonic compression test; (**a**) B1 and (**b**) B2 wallettes.

**Figure 9 polymers-14-01800-f009:**
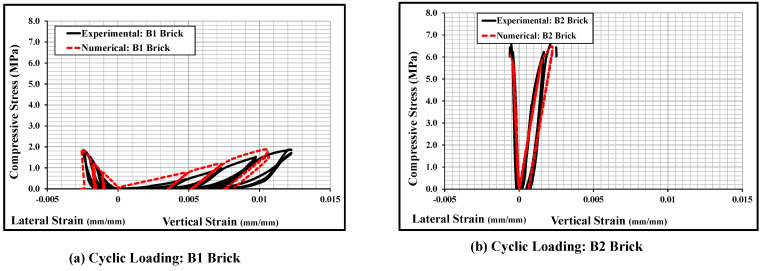
Stress-strain responses of the wallettes under cyclic compression test; (**a**) B1 and (**b**) B2 wallettes.

**Figure 10 polymers-14-01800-f010:**
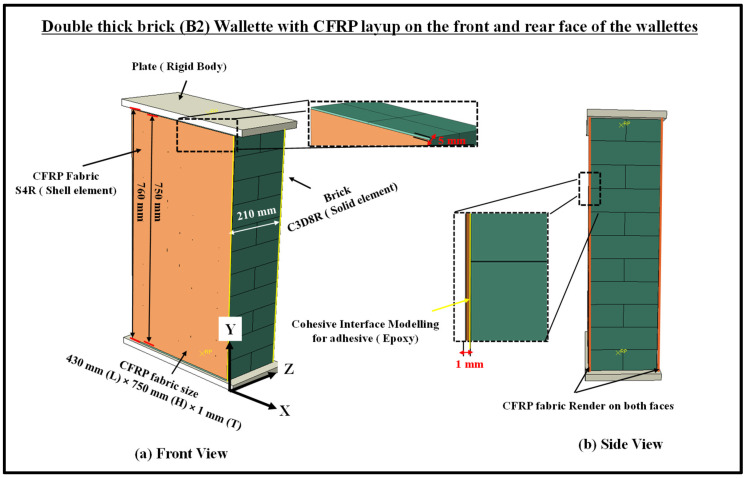
Numerical model of the rendering of CFRP/Auxetic fabric on the faces of masonry wallettes.

**Figure 11 polymers-14-01800-f011:**
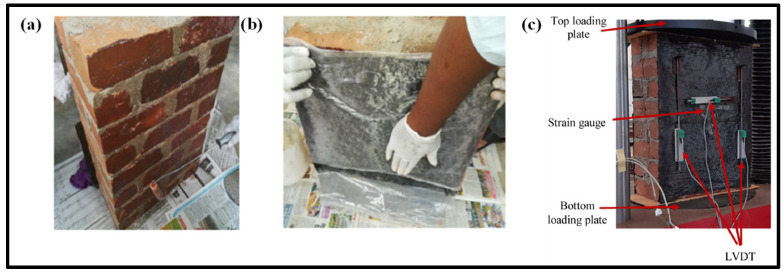
Preparation of CFRP-strengthened masonry wallettes; (**a**) bare masonry wallettes (**b**) CFRP layup using epoxy and (**c**) LVDTs attached on the surface of the CFRP fabric.

**Figure 12 polymers-14-01800-f012:**
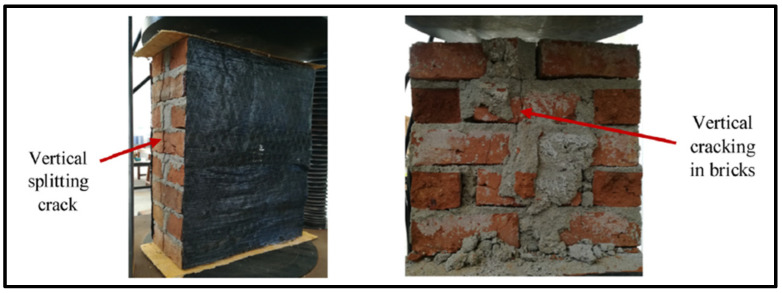
Failure modes of CFRP-strengthened masonry wallettes under monotonic loading (Reprinted with permission from [16]).

**Figure 13 polymers-14-01800-f013:**
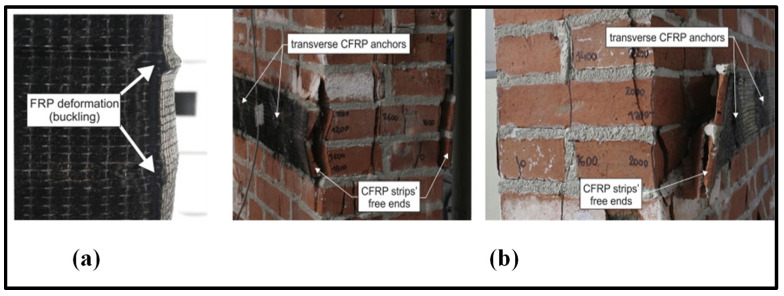
Failure modes on the CFRP-strengthened masonry wallettes (Adapted with permission from [59]); (**a**) CFRP deformation buckling at the edges and (**b**) premature interaction failure initiated at the edges.

**Figure 14 polymers-14-01800-f014:**
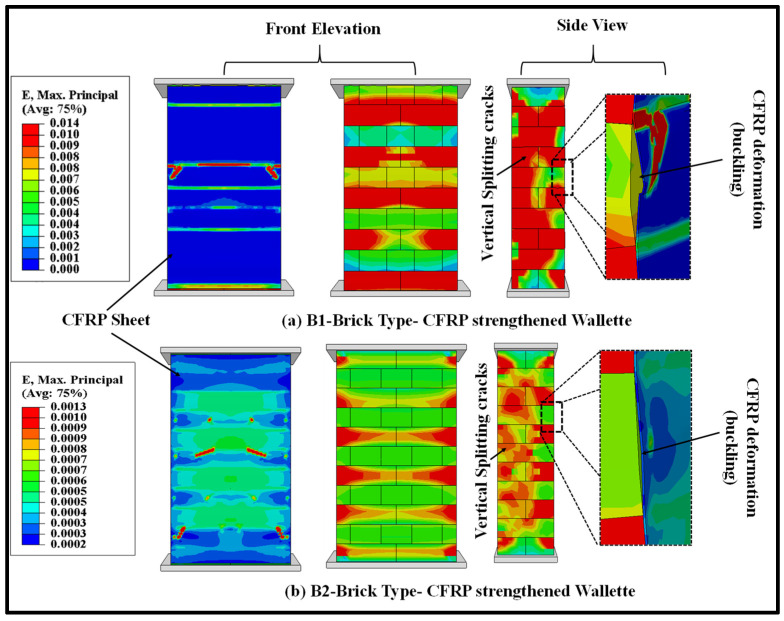
Numerically predicted damage on the surface of CFRP fabric and masonry wallettes constructed of different brick types; (**a**) B1 and (**b**) B2 under monotonic loading.

**Figure 15 polymers-14-01800-f015:**
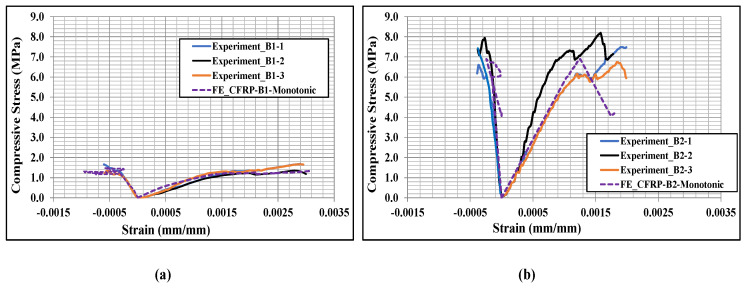
Stress-strain responses of CFRP-strengthened masonry wallettes constructed of different types of brick; (**a**) B1 and (**b**) B2 under monotonic loading.

**Figure 16 polymers-14-01800-f016:**
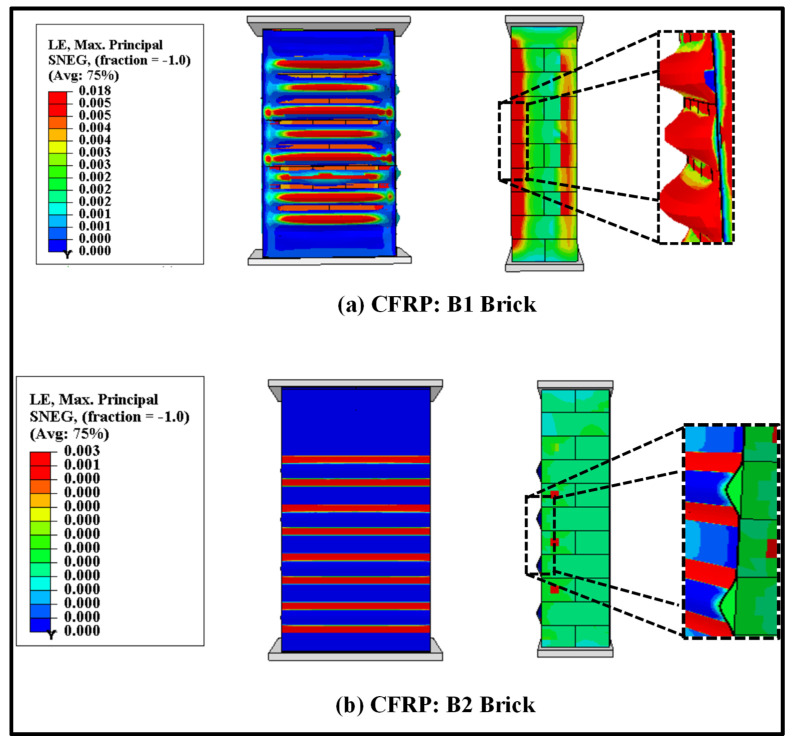
Numerically predicted damage on the surface of the CFRP fabric and masonry wallettes constructed of different brick types; (**a**) B1 and (**b**) B2 under cyclic loading.

**Figure 17 polymers-14-01800-f017:**
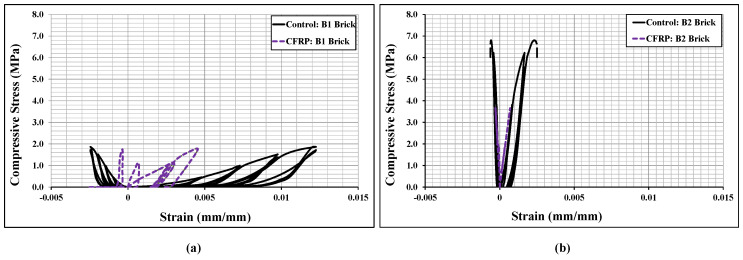
Stress-strain curves of the CFRP-strengthened cyclic loaded wallettes constructed using (**a**) B1 and (**b**) B2 Bricks tested under cyclic loading.

**Figure 18 polymers-14-01800-f018:**
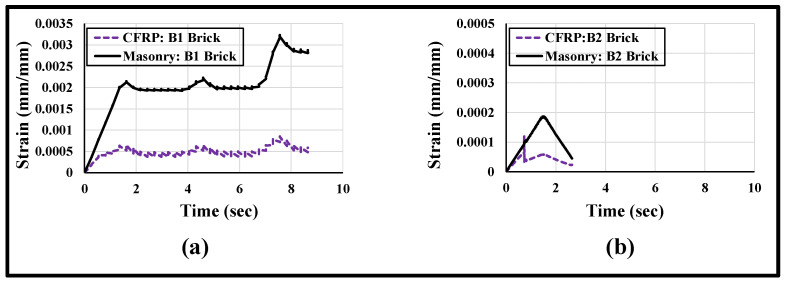
Strain history curve (numerically predicted) plotted at mid-height on the surface of CFRP and masonry wallettes constructed of different brick types, (**a**) B1 and (**b**) B2, tested under cyclic loading.

**Figure 19 polymers-14-01800-f019:**
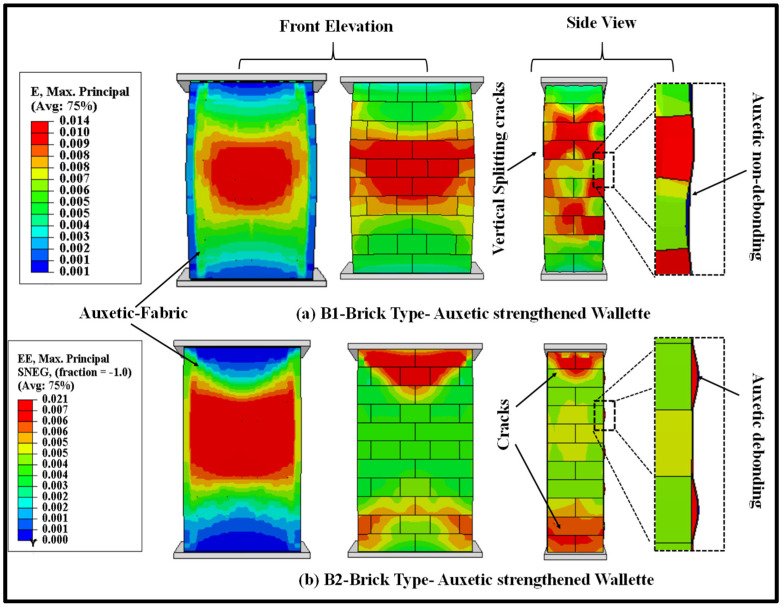
Numerically predicted damage on the surface of the auxetic fabric and masonry wallettes constructed of different brick types, (**a**) B1 and (**b**) B2, under monotonic loading.

**Figure 20 polymers-14-01800-f020:**
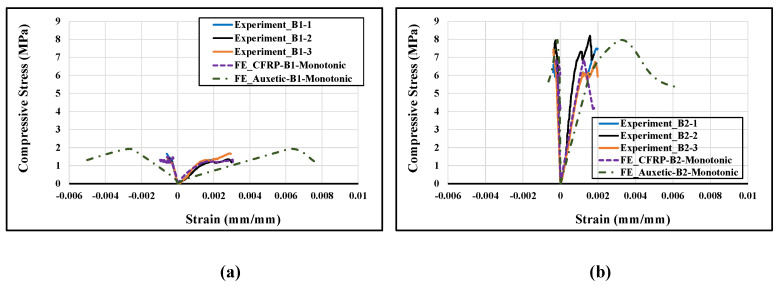
Stress-strain curve of CFRP-strengthened masonry wallettes construct of different types of bricks (**a**) B1 and (**b**) B2, under monotonic loading.

**Figure 21 polymers-14-01800-f021:**
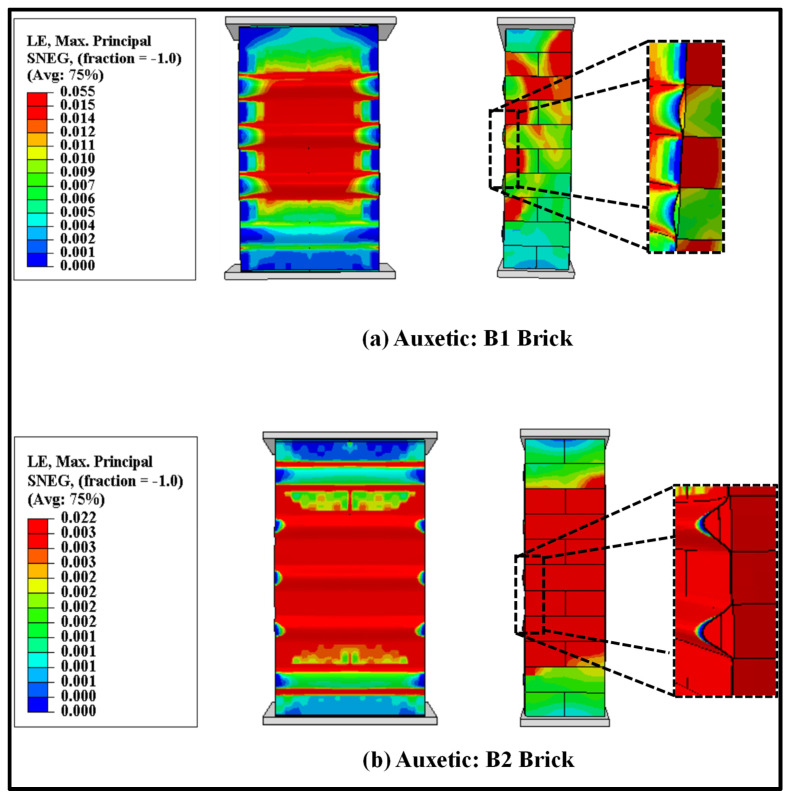
Numerically predicted damage on the surface of the auxetic fabric and masonry wallettes constructed of different brick types, (**a**) B1 and (**b**) B2, tested under cyclic compression loading.

**Figure 22 polymers-14-01800-f022:**
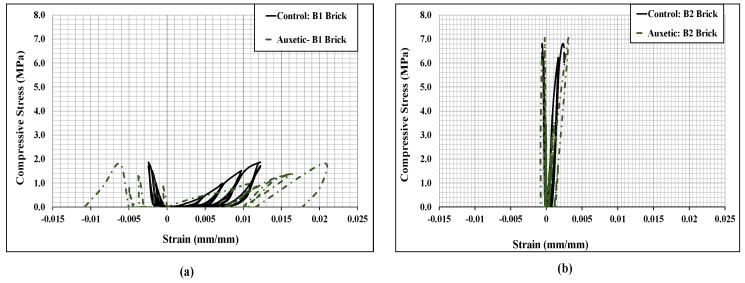
Stress-strain curves of the auxetic-strengthened cyclic loaded wallettes constructed using (**a**) B1 and (**b**) B2 bricks under cyclic compression loading.

**Figure 23 polymers-14-01800-f023:**
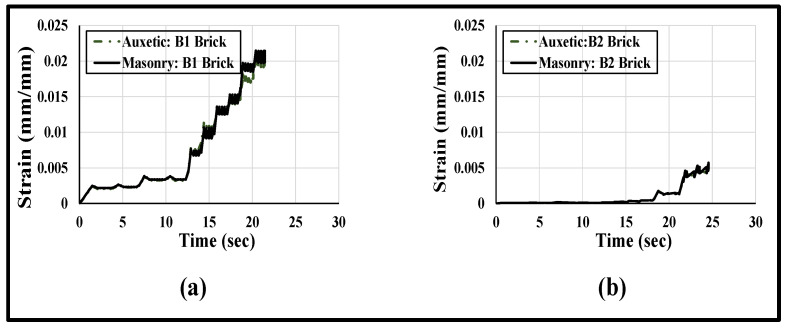
Strain history curve (numerically predicted) plotted at mid-height on the surface of auxetic fabric and masonry wallettes constructed of different brick types, (**a**) B1 and (**b**) B2, under cyclic compression loading.

**Table 1 polymers-14-01800-t001:** Dimensions and observed compressive strength of test specimens (Adapted from [16]).

Wallettes	Wallette Dimension (mm) (L × W × H)	Type of Test	Number of Wallettes Tested	Compressive Strength MPa (CoV)
B1 Brick	410 × 200 × 740	Monotonic Cyclic	4 3	2.28 (9.3) 1.93 (12.7)
B2 Brick	410 × 200 × 740	Monotonic Cyclic	4 3	6.66 (5.4) 5.46 (6.5)

**Table 2 polymers-14-01800-t002:** Cyclic loading protocol derived from monotonic loading (Adapted from [16,17,64]).

Monotonic Loading	Cyclic Loading	
Load-Displacement Response	Number of Steps	Number of Cycles
Elastic Limit (one-third of peak load)	4 steps	2 cycles at each step
Hardening Limit (0.8 times the peak load)	4 Steps	2 cycles at each step
Peak Limit	3 Steps	2 cycles at each step

**Table 3 polymers-14-01800-t003:** CDP model parameters.

Parameter	Magnitude
Dilation angle (*ψ*)	300
Eccentricity (*e*)	0.1
Strength ratio (*f_b_*_0_*/f_c_*_0_)	1.16
Shape factor (*K_c_*)	0.66
Viscosity (*µ*)	0.001

**Table 4 polymers-14-01800-t004:** Mechanical properties of the cohesive interfaces.

Parameter	Magnitude
Normal stiffness, $K_{nn}$ (N/mm^3^)	28
Shear stiffness, $K_{ss}\;\mathrm{and}\;K_{tt}$ (N/mm^3^)	32
Friction coefficient	0.6
Maximum tensile stress, $\sigma_{n}^{o}$ (MPa)	0.68
Maximum shear stress, $\tau_{s}^{o}$ and $\tau_{t}^{o}$ (MPa)	0.82

**Table 5 polymers-14-01800-t005:** Elastic properties of CFRP fabric.

Young’s Modulus (*x*-Axis) *E*_1_ (MPa)	Young’s Modulus (*y*-Axis) *E*_2_ (MPa)	Poisson’s Ratio in (*xy* Plane) $\upsilon_{12}$	Shear Modulus (*xy* Plane) *G*_12_ (MPa)	Shear Modulus (*xz* Plane) *G*_13_ (MPa)	Shear Modulus (*yz* Plane) *G*_23_ (MPa)	Density (kg/m^3^)
88,600	22,200	0.33	7067	3000	3000	1820

**Table 6 polymers-14-01800-t006:** Strength of CFRP fabric.

Fibre Tensile Strength $X^{T}$ (MPa)	Tensile Strength (*y*-Axis) $Y^{T}$ (MPa)	Longitudinal Shear Strength $S^{L}$ (MPa)	Transverse Shear Strength $S^{T}$ (MPa)
903	150	40	10

**Table 7 polymers-14-01800-t007:** Fracture toughness of CFRP fabric.

Fibre Tensile Fracture Energy $G_{ft}$ (mJ/mm^2^)	Matrix Tensile Energy $G_{mt}$ (mJ/mm^2^)
91.6	0.22

**Table 8 polymers-14-01800-t008:** Input material properties of adhesive (epoxy) (Adapted from [78,79]).

Parameter	Value
Elastic modulus of adhesive, *E_a_*	1.995 GPa
Maximum tensile stress, $\sigma_{n,max}^{}$	49.3 MPa
Maximum shear stress, $\tau_{s,max}^{}$	44.4 MPa
Normal stiffness, $K_{nn}$	1.995 × 10^3^ N/m^3^
Shear stiffness, $K_{ss}=K_{tt}$	1 × 10^3^ N/m^3^
Maximum fracture energies in normal, $G_{t}^{c}$	3900 N/m
Maximum fracture energies in shear, $G_{s}^{c}$	110,000 N/m

**Table 9 polymers-14-01800-t009:** Elastic properties of auxetic fabric.

Young’s Modulus (*x*-Axis)/*E*_1_ (MPa)	Young’s Modulus (*y*-axis) *E*_2_ (MPa)	**Poisson’s Ratio in (*xy* Plane)** $\upsilon_{12}$	Shear Modulus (*xy* Plane) *G*_12_ (MPa)	Shear Modulus (*xz* Plane) *G*_13_ (MPa)	Shear Modulus (*yz* Plane) *G*_23_ (MPa)	Density (kg/m^3^)
400	181.2	−0.9	194.5	194.5	103.4	450

**Table 10 polymers-14-01800-t010:** Strength of auxetic fabric.

Fibre Tensile Strength $X^{T}$ (MPa)	Tensile Strength (*y*-Axis) $Y^{T}$ (MPa)	Longitudinal Shear Strength $S^{L}$ (MPa)	Transverse Shear Strength $S^{T}$ (MPa)
50.0	40.4	10.1	10.1

**Table 11 polymers-14-01800-t011:** Fracture toughness of auxetic fabric.

Fibre Tensile Fracture Energy $G_{ft}.\;(\mathrm{mJ}/{\mathrm{mm}}^{2})$	Matrix Tensile Energy $G_{mt}\;(\mathrm{mJ}/{\mathrm{mm}}^{2})$
130.3	130.3

## Data Availability

The raw/processed data required to reproduce these findings cannot be shared at this time as the data also forms part of an ongoing study.

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
