# Peer review of "The Effectiveness of CFRP- and Auxetic Fabric-Strengthened Brick Masonry under Axial Compression: A Numerical Investigation"

_polymers, 2022, doi:10.3390/polym14091800_

Round 1
Reviewer 2 Report
This paper deals with the effectiveness of CFRP and auxetic fabric strengthened brick masonry under axial compression. A lot of simulation work has been carried out, and is further verified by the experiment. The prediction of cyclic behaviour of CFRP fabric strengthened brickwork wallets was conducted. Overall, the paper is well organized and written, including some significant information. The following comments can further improve the quality of the paper.
*Abstract
In the abstract section, please clarify the agreement between the model and the experimental results. Are the experiments results derived from the other references? It should be further clarified. In addition, some quantitative index and analysis are suggested to add.
* Introduction
#1 The reviewer suggested that the authors add the evaluation on FRP type and its effects of the strengthening of masonry structures in terms of material cost, bearing capacity, service life and other factors. For example, the cost of carbon fiber is high, but its mechanical, fatigue properties and durability are more excellent. In contrast, the cost of glass fiber is relatively low. However, its mechanical properties, fatigue resistance and durability are relatively poor. Therefore, the influence of FRP type on the strengthening of masonry structures is different. Please see the latest research work on the cost and performance of CFRP and GFRP. Composite Structures, 2022. 281: 115060. Construction and Building Materials, 2022, 315: 125710. After summarizing the above work, the authors should clarify why the current paper adopts CFRP as strengthening materials.
2# The strengthening method of FRP to masonry structure should be further clarified. Is it externally bonded or prestressed strengthening techniques? For the former, 20–30% of the tensile strength of the FRPs can be utilized owing to the interface debonding failure. In contrast, prestressed strengthening technique may be a more effective method owing to the anchoring effect at both ends of FRP. Therefore, the effects of different strengthening methods should be simply summarized, for example Materials and Structures, 2018, 51:162.
3# [44-47] provides the simulations of masonry structures strengthened with FRP. Therefore, what is the relationship between the simulation work in this paper and the above work? The contribution and innovation of this paper should be further emphasized.
Results and Discussion
1# When comparing the experimental results and finite element simulation as shown in Fig. 6 and Fig. 7, it’s hard to be detect the fitting degree of the two cases. The corresponding relationship between some hand drawn cracks and colored stress patterns is difficult to verify the authenticity.
2# Why are the elastic modulus and tensile strength of CFRP so low in Table 5 and table 6? For the general strengthening of engineering structures, the general tensile strength of CFRP is required to be more than 2100 MPa and ~160 GPa (elastic modulus).
3# The prediction of cyclic behaviour of CFRP fabric strengthened brickwork wallets was conducted in terms of the control condition. If the strengthened component experienced a long service environment (temperature-humidity alternating and dynamic/static loading), is the above prediction work still applicable?
4# The conclusions should be greatly condensed, including only 3 ~ 4 key points.
Round 2
Reviewer 2 Report
The authors have provided a comprehensive revision, and the quality of the paper has been greatly improved. Therefore, it is recommended to accept the current paper.